# Exploring the Role of Testosterone Replacement Therapy in Benign Prostatic Hyperplasia and Prostate Cancer: A Review of Safety

**André Rizzuti [1], Gustavo Stocker [2] and Heitor O. Santos [3],***

1 School of Medicine, Estácio de Sá University (UNESA), Rio de Janeiro 20261-060, RJ, Brazil; Rizzutiandre@gmail.com
2 School of Medicine, University Center Assis Gurgacz Foundation (FAG), Cascavel 85806-095, PR, Brazil; Gustavostockermed@gmail.com
3 School of Medicine, Federal University of Uberlandia (UFU), Uberlandia 38400-902, MG, Brazil
* Correspondence: heitoroliveirasantos@gmail.com

**Abstract:** Increased risk of prostate diseases triggered by testosterone replacement therapy (TRT) remains a worldwide concern. That said, we reviewed the safety of TRT in the spheres of benign prostatic hyperplasia (BPH) and prostate cancer (PCa), exploring clinical findings in this regard. Compelling evidence based on meta-analyses of randomized and observational studies indicates safety for TRT in patients suffering from prostate disorders such as BPH and PCa, at the same time improving lower tract urinary symptoms. Thus, the harmful relationship geared toward androgens and BPH seems to be overestimated as TRT has sufficient safety and, if properly prescribed, may counteract several metabolic problems. Even after PCa treatment, the benefits of TRT could outweigh the risk of recurrence, but further long-term randomized clinical trials are needed to elucidate unresolved questions.

**Keywords:** androgens; benign prostatic hyperplasia; prostate cancer; testosterone; testosterone replacement therapy





## 1. Introduction

Decline in serum testosterone (T) levels is an overt clinical issue, mainly in aging men, and therefore massive scientific effort has been made to better address pharmacological and nonpharmacological strategies in this setting [1–6]. The onset of T decline occurs markedly from the fifth decade of life [7]. Estimates account for 20% of T deficiency among men aged 60–70 years, 30% among men aged 70–80 years, and 50% among men aged over 80 years [7]. Besides aging, other pathological conditions can lead to primary and secondary hypogonadism, such as infiltrative diseases, testicular and pituitary trauma, varicocele, cryptorchidism, orchitis, hyperprolactinemia, obesity, and some genetic syndromes [8].

Epidemiological studies show an association of male hypogonadism with a greater prevalence and risk of developing cardiometabolic disorders such as dyslipidemia, obesity, diabetes, and hypertension [9,10]. Several meta-analyses have shown that men with hypogonadism are at increased risk of all-cause mortality and cardiovascular diseases [7,8,11–13]. Testosterone replacement therapy (TRT), in turn, can ameliorate cardiometabolic parameters and reduce the risk of cardiovascular events [14,15].

Improved well-being and sexual desire are undisputable when male patients suffering from hypogonadism are on TRT [16]. Seemingly, TRT has no major implications for prostate-specific antigen (PSA) levels [17,18]; however, the side effects of TRT on prostate enlargement and related disorders remain a concern, particularly, benign prostatic hyperplasia (BPH) and prostate cancer (PCa), which are commonly responsible for urologist visits, so much so that PCa is the second most prevalent cancer and the fifth leading cause of

cancer-related mortality in males, mainly affecting middle-aged to elderly patients [19–21]. Thus, we performed a narrative review in an attempt to ascertain the clinical magnitude of TRT on BPH and PCa, with the principal focus on safety.

## 2. Material and Methods

A brief literature search was performed employing Medline, Embase, Scopus, Cochrane Library, and Web of Science from inception to December 2021. The following keywords were used: "hypogonadism" or "testosterone" or "testosterone administration" or "testosterone replacement therapy" or "total testosterone" and "benign prostatic hyperplasia" or "prostate cancer" or "lower tract urinary symptoms". Then, the basic concepts and the clinical findings were discussed in order to unify traditional and recent evidence through a narrative review for practitioners and scientists in the area of urology.

## 3. Testosterone Action on Prostate Growth

T and its 5-alpha-reduced metabolite dihydrotestosterone (DHT) are the main androgens involved in prostate growth and maturation [22]. Although DHT is a vital hormone for embryological differentiation of the prostate during the intrauterine period, it can be considered a triggering factor for prostate disorders such as PCa and BPH [23]. Both PCa and BPH are linked to abnormal prostate enlargement, but with different pathophysiological features [24]. Commonly, the latter is characterized by nodules growing from the peripheral zone, while the latter consists of noncancerous nodules from the transitional zone [25,26].

## 4. Diagnosis and Treatment of Benign Prostatic Hyperplasia and Prostate Cancer

### 4.1. Benign Prostatic Hyperplasia

The diagnosis of BPH is made on the basis of clinical history and digital rectal examination, which is the first step to verify if there is any degree of abnormality in the dimensions of the prostate [27]. Digital rectal examination is indicated in the fifth decade of a man's life; however, the results are underestimated compared to transrectal ultrasound [28]. The American Urology Association Symptom Index (AUASI) questionnaire is another tool for monitoring BPH. It consists of seven questions that include the main symptoms related to BPH, with specific scores ranging from 0 to 5 in each section, with an overall maximum score of 35 points. Symptom severity can be classified as mild (0–7 points), moderate (8–19 points), and severe (20–35 points). The AUASI has a good correlation with the extent of the patient's urinary problems [29].

Lower urinary tract symptoms (LUTS) are a major concern in patients suffering from BPH by decreasing their health-related quality of life [30]. LUTS are divided into obstructive and irritative symptoms, which occur due to the mass effect and compression of adjacent structures such as nervous tissue and urethra [31]. Urethral compression impairs urination and may cause incontinence and bladder obstruction through ensuing progress [31].

The treatment of BPH encompasses the use of 5-alpha-reductase inhibitors (i.e., dutasteride and finasteride) and alpha-adrenergic agents (e.g., doxazosin) [32]. These pharmacological agents mitigate BPH progression while improving LUTS [33]. Along these lines, as DHT blockade is a therapeutic approach aimed at decreasing the size of prostate hyperplasia patterns, there is a caution for TRT in patients with BPH or an increased risk of BPH onset [34]. Although androgen deprivation in adulthood leads to involution of the prostate tissue, serum T and DHT concentrations do not appear to be higher in men who develop BPH [35,36]. For instance, in the Physicians' Health Study, 320 men surgically treated for BPH had similar serum T concentrations—collected up to nine years before BPH surgery—than 320 men who had not developed prostate diseases [37].

### 4.2. Prostate Cancer

PSA values above 4.0 ng/mL are widely used as the first indication of PCa, mainly when the abnormal result is double-checked [38]. Similarly to BPH, PSA levels are combined

with digital rectal examination and prostate imaging tests to screen for early PCa, which is generally asymptomatic [39]. Prostate biopsy, in turn, is the gold standard diagnostic technique for detecting PCa [40].

The Gleason scoring system is a pivotal component of PCa diagnosis and is often used to guide decision-making [41]. Such a scoring system quantifies the histological prostate pattern through five grades consisting of the sum of two numbers representing the Gleason grade of the predominant pattern added to the grade of the next most common pattern. A Gleason score less than 6 is classified as grade 1 and has the lowest clinical magnitude; grade 2 is composed of a score of 7 with the number 3 as a predominant pattern (3 + 4 = 7); a sum of 7 with the number 4 as a predominant pattern (4 + 3 = 7) is classified as grade 3; Gleason score > 7 with grades 4 and 5 involves poorly differentiated tumors and represents the worst prognosis [42,43].

The presence or absence of symptoms and other associated comorbidities, as well as patient preferences, are key factors in determining the treatment of PCa [44]. Treatment options for localized PCa according to cancer severity/risk group involve active surveillance, radical prostatectomy, and radiation therapy with or without androgen deprivation therapy [45].

## 5. Clinical Findings

### 5.1. Evidence for Testosterone Replacement Therapy in Benign Prostatic Hyperplasia and Prostate Cancer

Notwithstanding all the concerns about TRT and its suggested side effects in BPH and PCa, massive research brings important data that are in favor of patients suffering from prostate disorders. Evidence synthesis with and without a meta-analysis geared toward safety is collectively discussed in the following subsections.

### 5.1.1. Synthesis without a Meta-Analysis

A study that prospectively examined men with symptomatic late-onset hypogonadism revealed that TRT improved not only sexual function, but also bladder function by increasing its capacity and compliance while decreasing detrusor pressure at the maximum flow rate [46].

A study evaluating 30 patients who underwent supraphysiological doses (200 mg/wk, IM) of T for contraception purposes over an 18-month period (12 months of treatment plus 6 months after discontinuation) reported that, despite a sustained increase in serum levels of T, estradiol, and DHT during treatment, there was no detectable increase in prostate size on digital rectal examination or any significant change in PSA concentrations [47]. A small but significant increase ($14.3 \pm 2.0\%$) in the maximal prostate transverse area was noted in four men, whereas no alteration was observed in the other patients [47].

Of note, low T levels can be considered a prognostic marker for high-grade PCa, as proposed by a study that investigated individuals with a biopsy Gleason score of 8 or greater [48]. Correspondingly, lower T levels are observed in patients with advanced-stage PCa [49]. Taking into account the relevance of this laboratory relationship, the effects of TRT in patients with aggressive PCa are discussed below.

In an epidemiological study assessing 1181 men who received TRT after PCa diagnosis, such a hormone treatment was not associated with increased overall or cancer-specific mortality [50]. In a retrospective cohort study of male veterans (40 to 89 years) addressing 313 patients with aggressive PCa on TRT, when compared to 190 untreated men, TRT was not associated with incident aggressive PCa (hazard ratio (HR), 0.89; 95% CI, 0.70–1.13) or any PCa (HR, 0.90; 95% CI, 0.81–1.01) when fully adjusted (age, race, hospitalization during the year prior to cohort entry, geography, BMI, medical comorbidities, repeated T and PSA screening) [51].

To date, however, debate persists as to whether TRT is problematic or beneficial for PCa progression. For instance, from the 57 studies selected in a systematic review examining the effects of TRT on the risk of PCa, only six studies explored this issue, of which two did

not observe clinical PCa progression while the Gleason score increased in 8–15% of patients in the other studies [52].

5.1.2. Meta-Analyses

A meta-analysis of randomized controlled trials (RCTs) in men who underwent TRT reported that the combined rate of all prostate events was significantly higher in T-treated men than in the placebo group (odds ratio (OR) = 1.78; 95% CI, 1.07–2.95). Rates of PCa, PSA > 4 ng/mL, and prostate biopsies were numerically higher in the T group than in the placebo group [53].

A meta-analysis of RCTs assessing the efficacy and safety of TRT in men with hypogonadism found an increase in prostate volume (median difference (MD), 1.58; 95% CI, 0.6–2.56; *p* = 0.002) after TRT, but PSA levels and International Prostate Symptom Scores (IPSS) did not change [54].

A meta-analysis of RCTs and non-RCTs (total of 51 studies) found no significant effect of TRT on the incidence of PCa or the need for prostate biopsy when compared with a placebo [55]. Furthermore, there was no significant difference between the groups in the risk of other prostatic and urological outcomes, such as PSA levels and changes in IPSS.

In a meta-analysis of 14 RCTs (n = 2029 subjects) examining TRT for late-onset hypogonadism, TRT did not worsen LUTS compared with a placebo [56]. There was no statistical significance for IPSS from the baseline to the TRT follow-up period compared to those who received a placebo irrespective of T formulations (topical, injectable, and oral administration) [56]. Moreover, a meta-analysis of 22 RCTs involving a total of 2351 patients found that TRT administered transdermally, orally, or by injection did not differ in the risk of PCa development; neither short-term (<12 months) nor long-term TRT (12–36 months) affected the PCa incidence [57].

A meta-analysis addressing 15 RCTs showed that TRT was not associated with a higher risk of developing PCa, although PSA levels rose and plateaued relative to the placebo groups [58].

Regarding meta-analyses of observational studies, one conducted to determine the relationship between TRT and the risk of recurrence in T-deficient survivors of curatively treated high-risk PCa showed that the biochemical recurrence (BCR) rate was lower than expected, thereby implying that TRT can have null effects for BCR risk [59]. In another meta-analysis including prospective and retrospective studies that evaluated hypogonadal patients with PCa, TRT after definitive PCa therapy was considered oncologically safe because it did not increase the BCR rate [60].

Taken together, the design, population, and main results of these meta-analyses are summarized in Table 1.

**Table 1.** Meta-analyses evaluating the risk of BPH and PCa development in patients undergoing TRT.

| Study | Design (Number of Studies) | Population | Route of Administration | Duration (Months) | Outcomes |
|---|---|---|---|---|---|
| Parizi et al., 2019 [60] | Prospective and retrospective (n = 21) | PCa men treated with TRT after definitive local therapy (n = 1084) | TD; IM; PO | 1–102 | TRT after definitive PCa therapy appears to be safe and does not increase the BCR rate |
| Teeling et al., 2018 [59] | Prospective and retrospective (n = 13) | High-risk PCa survivors undergoing TRT (n = 109) | TD patch; TD gel TU IM PO Pellet | 1–189 | No increased risk of BCR |

**Table 1.** *Cont.*

| Study | Design (Number of Studies) | Population | Route of Administration | Duration (Months) | Outcomes |
|---|---|---|---|---|---|
| Guo et al., 2016 [54] | RCTs (n = 16) | Men with T deficiency; comparison of TRT-treated and placebo-treated patients (n = 1921) | TE IM; TU, PO; TD gel; TU IM | 6–36 | IPSS did not change significantly in the TRT group. TRT also improved life quality, anthropometric status, and metabolic parameters |
| Kohn et al., 2016 [56] | RCTs (n = 14) | TRT for men with LOH and LUTS assessed (n = 2029) | TD patch; TD gel; TE IM; TU IM; TU, PO | 3–36 | No statistical significance for IPSS from the baseline to the end of the follow-up period in the men treated with TRT compared to the placebo group |
| Kang and Li, 2015 [58] | RCTs (n = 15) | Men undergoing TRT (n = 739) | TD patch; TD gel; TE IM; TU IM TU PO | 3–12 | TRT seemingly did not increase the risk of PCa |
| Cui et al., 2014 [57] | RCTs (n = 22) | Men undergoing TRT (n = 2351) | TD patch; TD gel; TE IM; TU IM TU, PO | 3–36 | TRT had short-term safety and did not promote PCa development or progression |
| Fernández-Balsells et al., 2010 [55] | RCTs and non-RCTs (n = 51) | Men with low or low-normal T levels and treated with any T formulation for at least three months (n = 2798) | TD patch; TD gel; TE IM; TU IM; TU, PO | 3–36 | There was no significant difference between TRT and the placebo in PSA levels and IPSS |
| Calof et al., 2005 [53] | RCTs (n = 19) | Men ≥ 45 years old with low or low-normal T levels (651 subjects treated with T and 433 subjects treated with a placebo) | TU IM; TE IM; TC IM; ME IM; scrotal patch; TU, PO; TD patch; TD gel | 3–36 | TRT in older men was associated with a significantly higher risk of prostate events than the placebo (OR = 1.78; 95% CI, 1.07–2.95). |

BCR, biochemical recurrence; BPH, benign prostatic hyperplasia; IM, intramuscularly; IPSS, International Prostate Symptom Score; LOH, late-onset hypogonadism; LUTS, lower urinary tract symptoms; ME, mixed esters; TC, testosterone cypionate; TD, transdermal; TE, testosterone enanthate; TRT, testosterone replacement therapy; TU, testosterone undecanoate; PCa, prostate cancer; PO, per os; PSA, prostate-specific antigen.

*5.2. Correlations between Androgen and Estrogen Levels with Benign Prostatic Hyperplasia and Prostate Cancer*

Not only androgens, but also a tight control of estrogens merits attention in men's health. For instance, yin and yang of estrogen screening are fundamental during TRT, as elevated estrogen levels in high responders to T aromatization as well as high dosages of aromatase inhibitors and selective estrogen receptor modulators (e.g., letrozole, anastrozole, tamoxifen, and raloxifene [61–63]) can be detrimental to the well-being and cardiometabolic parameters of TRT-treated patients [64,65].

It is no wonder that in the complex field of male urology, the simple management of estrogen levels, especially estradiol, has gained relevance for monitoring the burden of BPH. In a cross-sectional study consisting of older men (n = 320; median age = 61 years), upper versus lower estradiol tertiles (28 vs. 18 pg/mL) were associated with a 50% lower likelihood of having the AUASI > 7 (OR = 0.5; 95% CI, 0.3–0.9) [66].

In a case–control study (n = 708 cases and 709 controls) using data from the Prostate Cancer Prevention Trial, higher T and estradiol levels were associated with 36% (OR = 0.64;

95% CI, 0.43–0.95) and 28% (OR = 0.72; 95% CI, 0.53–0.98) lower likelihood of having BPH, respectively [67]. In addition, a higher T:17β-diol glucuronide ratio was associated with a 36% lower likelihood of having BPH (OR = 0.64; 95% CI, 0.46–0.89). In this sense, controls had higher T (521 ± 93 vs. 492 ± 186 ng/dl, $p = 0.005$), bioavailable T (305 ± 99 vs. 289 ± 94 ng/dl, $p = 0.002$), estradiol (32 ± 9 vs. 30 ± 9 pg/mL, $p = 0.035$), and 17β-diol glucuronide (104 ± 69 vs. 96 ± 70, $p = 0.036$) levels compared to cases. Although the research design does not infer causation, the null effect of higher endogenous T levels on BPH is a crucial finding to furnish common laboratory ranges without TRT interference. Equally important, the results collectively suggest decreased activity of 5-alpha-reductase in the pathophysiology of BPH, as 17β-diol glucuronide is an indirect measure of DHT formation.

New data suggest a role of estradiol in prostate carcinogenesis mediated by estrogen receptor subtypes which are highly expressed in prostasphere undifferentiated cells, indicating that prostate progenitor cells are a target for estrogen and its proliferative effects [68]. Other estrogen-related mechanisms have been proposed, e.g., direct mutagenic effects on DNA and epigenotoxicity, but the findings are incipient, and hence further studies are warranted [69].

Despite attractive mechanistic actions, no association was found between serum estrogen levels and PCa risk in a nested case–control study encompassing 1798 biopsy-proven PCa cases and 1798 matched controls [70].

## 6. Strengths and Limitations

This review can guide clinicians by summarizing the overall effects of TRT on BPH and PCa, shedding light on the safety of a disputable area in medicine related to widespread clinical beliefs and judgments without proper scientific scrutiny. Even providing a framework for clinicians, this article does not have the same significance as meta-analyses and evidence grade as guidelines, but at least unifies the compelling meta-analyses published thus far.

Intrinsic limitations are conceivable among the studies, as well as heterogeneity regarding the posology of TRT. For instance, while 75 to 100 mg/week or 150–200 mg every two weeks of intramuscular administration of T is the common therapeutic regimen, there are some nuances for pharmacokinetics and pharmacodynamics due to different esters (e.g., undecanoate, enanthate, cypionate, and propionate) [71–73]. Not surprisingly, controversies are more evident when embarking on other routes of administration (e.g., subcutaneous, topical, and oral) due to the discrepancy to reach stable T levels [74]. Despite the appealing wide spread of topical T gel, slight and great differences of T concentrations across gel formulations and their bioavailability depending on the topical application sites can affect the medical approach when particular pharmacological facets are neglected [75,76].

Ultimately, we reiterate that complementary literature is mandatory for clinicians who deal with TRT prescription so that dosing regimens can be personalized. This review does not address recommendations, and thus guidelines from high-impact medical journals cannot be ruled out.

## 7. Conclusions

Low T levels and accompanying metabolic disorders appear to have a greater negative impact on BPH and PCa than the caveats for TRT, so much so that TRT has a reasonable safety margin and, at proper dosing regimens, may assist in metabolic regulation. The risks of TRT for prostate disorders such as BPH and PCa seem to be overestimated. Even after treatment for PCa, the benefits of TRT could outweigh the risk of recurrence; however, future long-term RCTs are needed to draw solid conclusions.

**Author Contributions:** Conceptualization, H.O.S. and A.R.; methodology, H.O.S. and A.R.; investigation, H.O.S., A.R. and G.S.; data curation, H.O.S.; writing—original draft preparation, A.R., G.S. and H.O.S.; writing—review and editing, H.O.S.; supervision, H.O.S.; project administration, H.O.S. All authors have read and agreed to the published version of the manuscript.

**Funding:** The authors were supported by Coordenação de Aperfeiçoamento de Pessoal de Nível Superior—Brazil (CAPES).

**Institutional Review Board Statement:** Not applicable.

**Informed Consent Statement:** Not applicable.

**Data Availability Statement:** Not applicable.

**Conflicts of Interest:** The authors declare no conflict of interest.

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
