# Peer review of "Exploring the Role of Testosterone Replacement Therapy in Benign Prostatic Hyperplasia and Prostate Cancer: A Review of Safety"

_2673-4397, doi:10.3390/uro2010005_

Round 1

Reviewer 1 Report

Firstly, I would like to congratulate the authors for their work. The authors have reviewed the available literature on the role of Testosterone replacement therapy in BPH and Prostate cancer. The authors conclude that- Low T levels and accompanying metabolic disorders appear to have a greater negative impact on BPH and PCa than the caveats for TRT. 

The work is good and has merit. However, there are certain issues in the manuscript. My comments are as follows:

Title: The title needs to be changed: Exploring the role of .......................

Abstract: well-written. no changes are needed.

Introduction: Please add your hypothesis in 1-2 lines at the end of this section. Why you have reviewed the literature on this research topic?

-There are major grammatical errors. Please take the help of a Writing assistant or online software.

Basic concepts:

-Section 2.1 is very detailed and can be shortened extensively (not more than 3-4 lines). It has a lot of redundant information. There is no point in discussing basic information like location, the weight of the prostate, etc.

-Section 2.2: needs to be extensively reduced. Table 1 has no relation with this manuscript.

Clinical findings:

-The abbreviation used needs to be standardized—Cap or PCa

-Also, please correct the abbreviations in other sections of the manuscript.

-Please discuss the limitations of the study?

Overall, the authors have performed a narrative review. They have also compiled the information of published meta-analyses. But this study is underpowered. However, it will be better if they consider doing a bibliometric review. 

Author Response

Reviewer 1

Firstly, I would like to congratulate the authors for their work. The authors have reviewed the available literature on the role of Testosterone replacement therapy in BPH and Prostate cancer. The authors conclude that- Low T levels and accompanying metabolic disorders appear to have a greater negative impact on BPH and PCa than the caveats for TRT. 

The work is good and has merit. However, there are certain issues in the manuscript. My comments are as follows:

Title: The title needs to be changed: Exploring the role of .......................

Answer: Thank you very much for your feedback, we have adapted the title as requested.

Abstract: well-written. no changes are needed.

Introduction: Please add your hypothesis in 1-2 lines at the end of this section. Why you have reviewed the literature on this research topic?

Answer: The last paragraph of the introduction was reformulated arguing the reason that led us to revise the topic and conduct the study.

-There are major grammatical errors. Please take the help of a Writing assistant or online software.

Answer: Grammatical errors throughout the text have been carefully revised.

Basic concepts:

-Section 2.1 is very detailed and can be shortened extensively (not more than 3-4 lines). It has a lot of redundant information. There is no point in discussing basic information like location, the weight of the prostate, etc.

Answer: We have removed the excess of biological information while summarizing the physiological relationship between testosterone action on prostate growth in a short section (section 3.).

-Section 2.2: needs to be extensively reduced. Table 1 has no relation with this manuscript.

Answer: We have removed this part along with Table 1. Only concepts of diagnosis and treatment of prostate cancer and benign prostate hyperplasia were maintained before the principal findings in an attempt to offer a clearer writing transition.

Clinical findings:

-The abbreviation used needs to be standardized—Cap or PCa

Answer: We have reviewed the entire paper and all the abbreviations have been standardized to Prostate Cancer (PCa).

-Also, please correct the abbreviations in other sections of the manuscript.

Answer: All the abbreviations have been doubled-checked and standardized accordingly.

-Please discuss the limitations of the study.

Answer: We have created section 6 to discuss the limitations of our review as well as some caveats for the literature cited.

Overall, the authors have performed a narrative review. They have also compiled the information of published meta-analyses. But this study is underpowered. However, it will be better if they consider doing a bibliometric review. 

Answer: We have created a methods section (section 2) proper for narrative reviews, where the keywords and scientific database used were cited.

Reviewer 2 Report

The current study aims to evaluate the effects of testosterone replacement therapy (TRT) on benign prostatic hyperplasia (BPH) and prostate cancer (PCa) from physiological concepts to the most recent clinical evidence.

Authors reviewed current literature in order to provide a framework for clinicians.

The study demonstrates that TRT may assist in metabolic regulation, and the risk of leading to BPH and PCa (where the benefits could outweigh the risk of recurrence) is overestimated.

The authors should be congratulated for the work and for addressing an important topic. Only few points warrant mentions:

Major comment:

  1. Authors should clarify how the study was conducted. I suggest to create a new paragraph on materials and methods.
  2. Line 134 “it is shown that T is not a villain in BPH”, it’s a strong conclusion, but incomplete. Even if T is not a villain, it is not an ally. I suggest to ameliorate the sentence.
  3. Lines 144-147. I suggest to justify the paragraph, it seems to not reflect the aims of the study. To help, I suggest the reference “Crocetto, Felice et al. “Impact of Sexual Activity on the Risk of Male Genital Tumors: A Systematic Review of the Literature.” International journal of environmental research and public health 18,16 8500. 11 Aug. 2021, doi:10.3390/ijerph18168500”.
  4. Line 160. Authors reported data on PCa grading using TRT, are available data on staging?

Minor comments:

  1. In “Introduction” section, I suggest to report data on PSA, as it has an important role in both pathologies, so in the manuscript.
  2. Line 115-116: authors should indicate how AUASI is useful for the assessment of LUTS.
  3. Lines 122-128 are a copy of lines 113-119, please, edit this error.

Author Response

Reviewer 2

The current study aims to evaluate the effects of testosterone replacement therapy (TRT) on benign prostatic hyperplasia (BPH) and prostate cancer (PCa) from physiological concepts to the most recent clinical evidence.

Authors reviewed current literature in order to provide a framework for clinicians.

The study demonstrates that TRT may assist in metabolic regulation, and the risk of leading to BPH and PCa (where the benefits could outweigh the risk of recurrence) is overestimated.

The authors should be congratulated for the work and for addressing an important topic. Only few points warrant mentions:

Major comment:

  1. Authors should clarify how the study was conducted. I suggest to create a new paragraph on materials and methods.

Answer: Dear reviewer, thank you very much for your comments. Regarding the first comment, the section Material and Methods has been created as a means of describing how the narrative review was conducted

  1. Line 134 “it is shown that T is not a villain in BPH”, it’s a strong conclusion, but incomplete. Even if T is not a villain, it is not an ally. I suggest to ameliorate the sentence.

Answer: We have ameliorated this sentence by excluding the affirmative sentence and making a more neutral conclusion based on the current evidence.

  1. Lines 144-147. I suggest to justify the paragraph, it seems to not reflect the aims of the study. To help, I suggest the reference “Crocetto, Felice et al. “Impact of Sexual Activity on the Risk of Male Genital Tumors: A Systematic Review of the Literature.” International journal of environmental research and public health 18,16 8500. 11 Aug. 2021, doi:10.3390/ijerph18168500”.

Answer: We have removed this part and so it was necessary to add your suggested reference. At best, we have learned about the content explored in your paper suggestion. Thank you for sharing this paper.

  1. Line 160. Authors reported data on PCa grading using TRT, are available data on staging?

Answer:  We have added new papers in which different stages of PCa were emphasized (please see section 5.1.1). Overall, we have focused on high-grade PCa/advanced-stage PCa due to greater relevance.

Minor comments:

  1. In “Introduction” section, I suggest to report data on PSA, as it has an important role in both pathologies, so in the manuscript.

Answer: We have briefly cited PSA levels in the introduction and better discussed its reference range and its clinical use in section 4.2.

  1. Line 115-116: authors should indicate how AUAI is useful for the assessment of LUTS.

Answer: We have better discussed how AUASI and LUTS works and their mutual importance in patients who suffer from BPH (please, see section 4.1.)

  1. Lines 122-128 are a copy of lines 113-119, please, edit this error.

Answer: We have corrected this mistake.

Round 2

Reviewer 1 Report

I would like to thank the authors for revising the manuscript. All my comments have been addressed in the revised manuscript. The overall scientific quality of the manuscript has improved significantly.

Reviewer 2 Report

The authors responded to the requests